# Strategies and Methods for Improving the Efficiency of CRISPR/Cas9 Gene Editing in Plant Molecular Breeding

**DOI:** 10.3390/plants12071478

**Published:** 2023-03-28

**Authors:** Junming Zhou, Xinchao Luan, Yixuan Liu, Lixue Wang, Jiaxin Wang, Songnan Yang, Shuying Liu, Jun Zhang, Huijing Liu, Dan Yao

**Affiliations:** 1College of Life Sciences, Jilin Agricultural University, Changchun 130118, China; 20211241@mails.jlau.edu.cn (J.Z.); 20211240@mails.jlau.edu.com (X.L.); 20220842@mails.jlau.edu.cn (Y.L.); 20220850@mails.jlau.edu.com (L.W.); 1911200221@mails.jlau.edu.com (J.W.); liushuyingbr@jlau.edu.cn (S.L.); 2College of Agronomy, Jilin Agricultural University, Changchun 130118, China; soy@jlau.edu.cn (S.Y.); zhangjun@jlau.edu.cn (J.Z.)

**Keywords:** CRISPR/Cas9, gene editing, plant molecular breeding, SgRNA, PAM

## Abstract

Following recent developments and refinement, CRISPR-Cas9 gene-editing technology has become increasingly mature and is being widely used for crop improvement. The application of CRISPR/Cas9 enables the generation of transgene-free genome-edited plants in a short period and has the advantages of simplicity, high efficiency, high specificity, and low production costs, which greatly facilitate the study of gene functions. In plant molecular breeding, the gene-editing efficiency of the CRISPR-Cas9 system has proven to be a key step in influencing the effectiveness of molecular breeding, with improvements in gene-editing efficiency recently becoming a focus of reported scientific research. This review details strategies and methods for improving the efficiency of CRISPR/Cas9 gene editing in plant molecular breeding, including Cas9 variant enzyme engineering, the effect of multiple promoter driven Cas9, and gRNA efficient optimization and expression strategies. It also briefly introduces the optimization strategies of the CRISPR/Cas12a system and the application of BE and PE precision editing. These strategies are beneficial for the further development and optimization of gene editing systems in the field of plant molecular breeding.

## 1. Introduction

In 2002, a new family of DNA sequences found only in bacteria and archaea was discovered through bioinformatic analysis, they called this sequence ‘clustered regularly interspaced short palindromic repeats’ (CRISPR) and named the genes close to the CRISPR locus ‘Cas’ (CRISPR-associated) [1]. In 2012, the working principle of the 28 CRISPR/Cas9 gene editing technology was successfully elucidated [2]. In 2013, CRISPR/Cas9 was used for the first time in several fields to achieve not only a significant increase in gene knockout efficiency, but also to enable multiple gene knockouts [3,4,5,6,7]. In particular, in recent years, genetic control of plant genomes has been achieved through the use of CRISPR/Cas9 gene-editing technology for the genetic improvement of crops. CRISPR/Cas9 gene-editing technology has greatly advanced the process of molecular breeding and has revolutionized the field of gene-editing breeding [8,9,10,11,12,13].

Compared with conventional zinc-finger nucleases (ZFNs) and transcription activator-like effector nucleases (TALENs), CRISPR/Cas9 gene-editing technology has significant advantages in terms of gene editing capabilities and convenience. ZFNs and TALENs, of which *Fok* I (Type IIS restriction enzyme of *Flavobacterium okeanokoites*) nucleases are the main functional type and require dimerization of the *Fok* I nuclease structural domain to become active [14,15,16,17]. Cas9 requires only a short single guide RNA (SgRNA) with a 20 bp sequence to guide onto the genome and does not require nuclease modification [18,19,20]. In the CRISPR/Cas9 system, the Cas nuclease induces a double-strand breakage (DSB) at the designated target site of the gRNA [21]. The target DNA site (usually 20 nucleotides long) is often referred to as the original spacer sequence. Endogenous non-homologous end joining (NHEJ) and homology-directed repair (HDR) pathways repair DSBs [22,23,24,25]. While NHEJ is an error-prone repair process that often leads to the introduction of mutations such as minor insertions and deletions (Indels), HDR leads to the precise repair of DSBs. A common result of DSBs in the genome is the generation of random insertions or deletions through NHEJ, which is the predominant DSB repair pathway in plants [26,27,28,29].

CRISPR/Cas9 gene-editing technology has recently made some progress in the direction of plant molecular breeding, but still presents some problems that affect the process of technological development. For example, the expression of multiple gRNAs in an editing vector can affect the efficiency of gene editing [30,31,32], PAM (protospacer adjacent motif)-restriction recognition sequences [33,34], genetic instability, and off-target problems [35,36,37], etc. Much research and exploration in optimizing CRISPR/Cas9 gene-editing technology have been conducted. Scientists have sought to enhance gRNA expression by manipulating the primary elements of the gRNA expression process, such as the type of promoter used, the approach to multiple gRNAs, and the system of delivering CRISPR reagents to the target cells and tissues, thereby augmenting the editing proficiency of CRISPR/Cas9 gene-editing technology in the plant genome [38,39,40].

This review details strategies and methods of improving the efficiency of CRISPR/Cas9 gene editing in plant molecular breeding in recent years; it also considers the application and technical optimization of other interesting gene-editing technologies such as CRISPR/Cas12, BEs, and PEs in crop improvement.

## 2. Advances in CRISPR/Cas9 Gene Editing Technology in the Field of Plant Molecular Breeding

Following the successful application of the CRISPR/Cas9 system in human cells [3], researchers have successfully applied it to a variety of model crops such as *Arabidopsis* (*Arabidopsis thaliana* (L.) Heynh.) and tobacco (*Nicotiana tabacum* L.), creating new plant traits that show a wealth of promise. Shortly thereafter, various CRISPR/Cas9 vector systems were developed for more efficient editing of plant genomes, including knockouts, genomic deletions, disruption of cis-regulatory elements, gene insertion, and suppression of viral infections [41,42,43,44].

In 2013, Shan et al. successfully used CRISPR/Cas9 gene-editing technology to achieve the knockdown of *AtPDS3* and *NbPDS* genes in *Arabidopsis* and tobacco, respectively. This created phenotypes that abolish carotenoid biosynthesis and promote chlorophyll oxidation, leading to a photobleached phenotype, thereby achieving the first application of CRISPR/Cas9 technology in plant genomes [45]. In 2013, Nekrasov et al. developed a novel transient transformation method based on the CRISPR/Cas9 gene editing system in tobacco, accelerating the development of CRISPR/Cas9 gene editing systems for plant genome applications [46]. In 2013, Li et al. successfully knocked out *OsPDS-SP1* and *OsBADH2* in the rice (*Oryza sativa* L.) genome, creating the first transgenic rice with albino and dwarf phenotypes [47]. In the early days, the editing of plant genomes was limited to single genes or single targets, and the editing efficiency was affected by many factors such as the limitations of PAM selection, Cas9 fidelity, and off-target phenomena, which affected the application and development of CRISPR/Cas9 [48,49,50,51,52,53].

Researchers have adapted and improved the CRISPR/Cas9 gene-editing technology, which is now employed in a variety of plants, such as maize (*Zea mays* L.), wheat (*Triticum aestivum* L.), banana (*Musa* sp.), apple (*Malus pumila* Mill.), tomato (*Solanum lycopersicum L.*), soybean (*Glycine max* (L.) Merr.), and rice (Figure 1) [54,55,56]. Its intended applications include increasing crop yields, improving crop quality traits, enhancing resistance to abiotic or biotic stresses, etc. [57,58,59]. Compared with its immaturity in early applications, CRISPR/Cas9 is striking for its powerful gene-editing efficiency and ease of use [60]. In the last three years of reported research, and due to the functional redundancy of most plant genes and their complex interactions, scientists have addressed target genes in the form of multiple gene editing, simultaneously enabling the editing of multiple target loci in the genome; this has greatly reduced the cycle time and difficulty of obtaining higher order mutants in research [61,62].

In 2022, using a single gRNA CRISPR/Cas9 gene editing process, Zhang et al. knocked out three soybean PYL genes: *GmPYL17*, *GmPYL18*, and *GmPYL19*; they verified that the polygenic mutants were less sensitive to ABA and had higher mutant plant height and branch number than the wild type [63]. In 2022, Liu et al. edited several male sterility genes in maize, simultaneously confirming the gene functions of each member of the *ZmTGA9* family; they identified *ZmDFR1*, *ZmDFR2*, *ZmACOS5-1*, and *ZmACOS5-2* as controlling male fertility in maize [64]. In 2022, investigating rice, Fu et al. created a multi-gene mutant of the rice *Wx* gene family and demonstrated that the WX mutation significantly reduced AAC and starch viscosity but did not affect major agronomic traits [65]. In 2023, working on wheat, Kan et al. created a series of *TaeIF4E* single, double, and triple knockout mutant resources. They found that only the *TaeIF4E* triple mutant was completely resistant to WYMV and set fruit normally; the single or double mutant remained susceptible and showed dwarf plants and severely reduced fruit set after susceptibility [66]. Not only has multi-gene editing been used on a large scale in major crops, but new superior traits have been created through multi-gene editing in a variety of crops such as tomatoes, apples, bananas, and potatoes (*Solanum tuberosum* L.) [67,68,69,70,71,72].

Studying multiple related genes, knocking out functionally redundant genes, and genetic improvement of multiple traits in crop breeding render simultaneous editing of multiple genomic loci in plants a major means and method for future plant molecular breeding. This improves the gene-editing efficiency of CRISPR/Cas9 gene-editing technology in plant molecular genetic breeding and is the key to creating plants with multiple superior traits [73,74,75,76].

## 3. Strategies and Methods for Optimizing CRISPR/Cas9 Gene-Editing Technology

In the context of the widespread use of CRISPR/Cas9 multiple gene editing systems in plant genomes, improving the efficiency of gene editing has recently become a major research direction in recent scientific reports. In this section, we detail the methods and applications of Cas9 variant enzyme engineering, the effect of multiple promoter driven Cas9 and gRNA efficient optimization, and expression strategies to improve the efficiency of gene editing.

### 3.1. Development of Cas9 Variant Enzymes to Expand Recognition of PAM and Improve Editing Efficiency

Cas9 and gRNA, as genome editing tools, must bind to a specific PAM sequence, a short nucleotide sequence located at the 3′ end of the target sequence, and PAM restriction recognition sequences remain one of the key issues hindering the development of CRISPR systems [77,78,79]. Scientists have developed different types of Cas9 variant enzymes to expand the range of PAM recognition to improve the efficiency of gene editing (Table 1). *Streptococcus pyogenes* Cas9 (SpCas9) was the first Cas9 protein to be used in plant genome editing and remains the most commonly used Cas9 protein in the CRISPR system for plant molecular breeding [80,81]. In the function of target sequence cleavage, SpCas9, CRISPR RNA (crRNA), and trans-activating CRISPR RNA (tracrRNA) combine to form ribonucleoprotein (RNP) complexes to participate in this process [82,83]. The SPCas9-gRNA complex normally recognizes a region 20 nt upstream of the PAM sequence (5′-NGG-3′). This means that sequences without NGG cannot be selected as target sequences [84,85]. Therefore, various methods have been used to expand the affinity and enhance the specificity of PAM. These include Rational SPCas9 engineering, identification and characterization of Cas9 homologs, and new CRISPR/Cas systems from other sources [86]. Based on the crystal structure of Cas9, rational modification engineering of the SPCas9 protein using sgRNA and target DNA can produce engineered Cas9 proteins with different PAM preferences [87,88,89]. New Cas9 proteins reported in recent years include ScCas9, SaCas9, StCas9, and CjCas9, which are variants of the enzymes found in different bacteria that recognize different PAM sequence ranges. New Cas9 proteins also include the branching variants Cas9, SpCas9-NG, SpCas9-VQR, SpCas9-EQR, fCas9, xCas9, etc., which can improve fidelity and have high editing efficiency. These variant enzymes have been applied to genome editing in model crops such as *Arabidopsis*, rice, and tobacco, significantly extending the scope of Cas9-mediated genome editing in plants and improving the efficiency of CRISPR gene editing. Novel Cas proteins such as HF-Cas9, HypaCas9, eSpCas9, and Sniper Cas9 also show significantly lower off-target levels when applied in plants [90,91,92,93,94].

In 2018, Zhang et al. used eight SpCas9 variant enzymes with improved tRNA-sgRNA fusions to significantly improve editing efficiency in plant genomes [95]. In 2019, Zhong et al. first reported the use of two variants of SpCas9, xCas9 and Cas9-NG, in the rice genome and found that they significantly improved the efficiency of gene editing [96]. In 2020, Zheng et al. significantly improved gene-editing efficiency using ubiquitin-related structural domains in combination with SpCas9 in rice protoplasts and stable transformation [42]. In 2021, Kurokawa et al. used the parsley UBIQITIN promoter to drive the SpCas9 protein with significantly improved gene-editing efficiency at all four target loci in the *Arabidopsis* genome [97]. In 2021, Carrijo et al. developed two editing systems based on SpCas9: STU and TCTU, which greatly improved the efficiency of editing in the soybean genome [98]. In 2021, Liu et al. reported that ScCas9(++), an improved enzyme of ScCas9, was edited more efficiently than ScCas9 in rice. They also reported the development of a new evoBE4max-type cytidine base editor which fused the evolved cytidine deaminase coding gene *PmCDA1* with ScCas9(++); this achieved stable and efficient multiple-site base editing at the NNG-PAM locus with a wide editing window for any target sequence [99]. In 2021, Li et al. found that zCas9 had a high editing efficiency for the *Rehmannia glutinosa* genome [100]. In 2022, Jedlickova et al. developed plant codon-optimized *Streptococcus pyogenes* Cas9 (pcoCas9), increasing the efficiency of gene editing by 25% [101].

Among the Cas9 variant enzymes, SpCas9-NG has the widest range of PAM sequences, recognizing NGD (containing NGG, NGA, and NGT), RGC (containing AGC and GGC), GAA, and GAT PAM sites [102,103]. To improve the efficiency of HDR-mediated editing, base editors that enable precise base editing have been developed. As some of the SpCas9 variant enzymes can recognize unconventional PAMs (e.g., XCas9 and SPCas9-NG), an increasing number of variant enzymes are being used in base editing [104,105,106]. In 2019, Xu et al. developed three high-fidelity SpCas9 variants, eSpCas9, SpCas9-HF2, and HypaCas9, designed to act as C-T base editors together with *PmCDA1* (pBEs). The knockout mutation frequency of these high-fidelity Cas9s was increased 2~5-fold, with eSpCas9(1.1)-pBE being 25.5-fold more efficient. In 2019, Wu et al. used SpCas9n and VQRn to achieve a 1.3~7.6-fold base-editing efficiency in the rice genome [107]. In 2022, Tan et al. developed adenine base editors (ABEs) based on the principle of modified adenosine deaminases and Cas variants that introduce site-specific A-to-G mutations for agronomic trait improvement. The SpCas9n-NG and SpGn used in the editing process recognize PAM: NGN or NNN and not only allow for an expanded target range and a wider editing window, but also have significantly improved base-editing activity [108]. In 2023, Qiao et al. added a structured RNA motif, evopreQ1, to the 3′ end of pegRNAs to improve their stability by preventing degradation and named the motif epegRNAs. They also optimized the PE2 protein to obtain PEmax and over-expressed the MLH1 protein with a disrupted active structural domain (MLH1dn) to inhibit the DNA mismatch repair pathway, thereby substantially improving the efficiency of PE editing in the maize genome [109].

**Table 1 plants-12-01478-t001:** Cas9 variant enzymes and the effects obtained in the plant genome.

Cas9 Nuclease	Origin	Identifying PAM	Cutting Activation	Improving the Efficiency of Gene Editing	References
SpCas9	*S. pyogenes*	NGGN	49%	80%	[108]
SaCas9	*S. aureus*	NNNRRT, NNGRRT	50%	60.6%	[110]
ScCas9	*S.canis*	NTG-, NGG-, NCG-	53.6%	57.2%	[111]
xCas9	*S. pyogenes*	NG, GAA, GAT	32%	21.1%	[112]
Cas9-NG	*S. pyogenes*	NG	30%	56.8%	[108]
eSpCas9	*S. pyogenes*	NGG	40%	80%	[113]
evoCas9	*S. pyogenes*	NGG	15%	-	[114]
SpCas9-HF2	*S. pyogenes*	NGG	34%	65%	[115]
Sniper Cas9	*S. pyogenes*	NGG	46%	-	[116]
HypaCas9	*S. pyogenes*	NGG	30%	-	[116]

### 3.2. Efficient Expression of Multiple sgRNAs Improves the Efficiency of Gene Editing

In recent years, researchers have increasingly reported the expression of multiple gRNAs targeting multiple target genes in plant editing vectors and single gRNA recognition sites containing two or more target genes. These are performed in order to obtain multiple gene-editing mutants with more pronounced changes in plant traits and to improve the efficiency of gene editing (Table 2) [117,118,119]. The gRNA is a small non-coding RNA, so its expression is usually initiated using the U3 or U6 promoter corresponding to the snoRNA. The transcripts of the U3 and U6 promoters must be expressed from nucleotides “A” and “G”, respectively, which greatly limits the choice of targeting sequences for the gene being edited. To achieve efficient gene editing, most researchers design their sgRNA editing sites to target multiple target genes, thereby obtaining multiple mutants to improve the efficiency of gene editing [120]. In 2020, Li et al. designed three gRNAs to target four *LNK2* genes in soybean to obtain a tetra-gene mutant [121]. In 2021, Wang et al. used three sgRNAs to target six *GmAITR* genes to obtain a five-gene mutant [122]. In 2022, when designing an editing vector to obtain a double mutant, Lu et al. designed one sgRNA to target two *GmPDS* genes [123]. By editing multiple functionally identical or similar target genes simultaneously, higher-order mutants undergo more pronounced changes in agronomic or quality traits, often creating unexpected breeding effects [124,125,126]. In 2018, Yu et al. designed single editing sites in the *Arabidopsis* genome to target three target genes, which not only improved the efficiency of gene editing but also greatly reduced the probability of off-targeting [127]. In 2019, Bao et al. constructed single and double gene-editing vectors to target the *GmSPL9a* and *GmSPL9b* genes, respectively; they obtained mutants and then performed phenotypic analysis to find that the double gene mutants were more pronounced in terms of phenotypic changes, while the single gene mutants showed no significant changes in terms of phenotype [128]. In 2021, Wang et al. found that only the *ZmBADH2a* and *ZmBADH2b* double mutants were phenotypically altered when they edited the betaine aldehyde dehydrogenase gene [129]. In 2021, Ren et al. were able to enhance the efficiency of CRISPR/Cas9 gene editing using the *VvU3* and *VvU6* promoters and two ubiquitin (UBQ) promoters in grapevine genome editing [130]. In 2022, Cao et al. designed sgRNA target loci to obtain *RS2* and *RS3* double mutants and *RS2* single mutants, and found that the double mutants were more pronounced in terms of phenotypic changes [131]. In 2022, Biswas et al. performed multiple CRISPR-Cas9 genome editing in peanut protoplasts and validated gRNA activity using a polyethylene glycol (PEG)-mediated protoplast transformation system [132]. In 2022, Zhang et al. used the MaU6c promoter to improve editing efficiency fourfold in banana protoplasts [133]. The most common method of expressing multiple gRNAs is to create expression frames containing multiple independent gRNAs through their promoters and terminators. In 2021, Kim et al. used a promoter, one sgRNA, and a multi-level tRNA-gRNA strategy to design multiple sgRNAs targeting multiple genes in the soybean *FAD2* and *FATB* gene families, enabling sgRNAs targeting different target genes to be assembled into the same vector and successfully edited [134].

In 2019, Bai et al. used a gRNA pooling strategy to design a total of 70 gRNAs targeting the *GmRIC* gene; they assembled two, three, four, and five gRNAs in the vector and found that the higher the number of gRNAs, the lower the efficiency of vector editing, often with only 2-4 gRNAs functioning [135]. From this, we deduce that in CRISPR/Cas9-mediated multiple-gene editing systems, the efficiency of gene editing decreases significantly when the number of gRNAs expressed simultaneously exceeds a certain number. When there are too many gRNAs, competition between them can lead to a reduction in the efficiency of gene editing. The efficiency of simultaneous editing of all genes is equal to the product of the efficiency of editing all individual genes [136]. If individual genes are edited too inefficiently, it makes later screening and identification doubly difficult and time-consuming, so it is also important not to express too many gRNAs when constructing editing vectors [137]. Although it is difficult to obtain higher-order mutants using a single editing vector, we can use crosses between stably inherited mutants of different genes to obtain multiple mutants [138,139,140]. In 2022, Mu et al. created two different double mutants by targeting two genes with a single sgRNA and then obtained *GmBIC* quadruple mutants by hybridization. In the future, obtaining multiple mutants more quickly and easily will be the main optimization direction of the CRISPR/Cas9 gene editing system [141].

**Table 2 plants-12-01478-t002:** CRISPR/Cas9-targeted genes and editing efficiency in major agricultural crops.

Species	Name of Gene	Transformation Method	Gene Function	Number of sgRNA	Gene Editing Efficiency	Acquisition of New Traits	References
Soybean	*GmFAD2*	*Agrobacterium*-mediated method	Regulation of oil synthesis	2	53.3%	Creating high oleic acid soybeans	[142]
	*GmIPK1*	*Agrobacterium*-mediated method	Regulation of phytic acid content	1	84.3%	Creating low phytic acid soybean seeds	[143]
	*GmPDH1*	*Agrobacterium*-mediated method	Regulating pod breakage	3	43.4%	Creating pods of unbreakable soybeans	[144]
	*GmPDS*	*Agrobacterium*-mediated method	Modulating the albino and dwarf phenotypes	2	87.5%	Soybean for the creation of dwarf and albino phenotypes	[145]
	*GmRS2,* *GmRS3*	*Agrobacterium*-mediated method	Regulation of oligosaccharide content in soybeans	2	50.5%	Creating low oligosaccharide soybeans	[146]
Rice	*OsPUT*	*Agrobacterium*-mediated method	Regulation of paraquat resistance in rice	3	75%	Creating glufosinate tolerant rice	[147]
	*OsWx*	*Agrobacterium*-mediated method	Regulation of starch content in rice	5	82.5%	Creating low starch content rice	[148]
	*OsDjA2, OsERF104*	Electroporation	Regulation of rice resistance to plague and blight	1	66.65%	Creation of plague-resistant rice	[149]
	*OsGlu*	*Agrobacterium*-mediated method	Regulation of protein content in rice	4	79.2%	Creation of high protein rice	[150]
	*OsSAP*	*Agrobacterium*-mediated method	Regulating drought tolerance in rice	1	43.2%	Creating drought-resistant rice	[151]
Maize	*ZmGDIα*	*Agrobacterium*-mediated method	Regulation of coarse and short traits in maize	1	40.98%	Creating coarse dwarf disease resistant maize	[152]
	*ZmAbh4*	*Agrobacterium*-mediated method	Regulation of maize water use efficiency	1	26.7%	Creation of high moisture utilisation maize	[153]
	*ZmTGA9*	*Agrobacterium*-mediated method	Regulation of male sterility traits	2	80%	Creation of male sterile maize	[68]
	*ZmFER1*	*Agrobacterium*-mediated method	Regulation of resistance to Fusarium spike rot	1	60%	Creation of Fusarium spike rot resistant maize	[154]
	*ZmMYB69*	*Agrobacterium*-mediated method	Regulation of lignin synthesis in maize	2	40%	Creation of lignin synthesis inhibiting maize	[155]

### 3.3. Exploring Efficient Promoter-Activated Expression of Cas9 to Improve Editing Efficiency

Promoters are key to driving the expression of transformed genes: in addition to the most used 35S promoter of the constitutive cauliflower mosaic virus (CaMV), various other promoters have been used in an attempt to express Cas9 and improve its editing efficiency [156]. In 2018, Hashimoto et al. used the tomato elongation factor-1α (SlEF1α) promoter to drive Cas9 to improve the efficiency of gene editing in the tomato genome [157]. In 2019, Bai et al. used the strong endogenous promoter Progmscream M4 to drive Cas9 [135]. In 2019, Kishi-Kaboshi et al. demonstrated better results than the CaMV35S promoter in chrysanthemum breeding for the first time using the parsley ubiquitin (PcUbi) promoter [158]. In 2020, Wolabu et al. compared the effects of four different promoters in *Arabidopsis* and found that Cas9 driven by the *Arabidopsis thaliana* UBQ10 promoter significantly increased the efficiency of gene editing by 95% [159]. In 2020, Li et al. compared four different RNA polymerase (Pol) III promoters (TaU3p, TaU6p, OsU3p, and OsU6p) in rice protoplast transformation and found that the optimized sgRNA scaffold driven by the TaU3 promoter was the most efficient [160]. In 2021, An et al. used the mannan synthase (MAS) promoter to drive Cas9, further increasing the mutation rate at the edit site by up to 75% [161]. In 2021, Massel et al. used the endogenous U6 promoter (SbU62.3) to improve CRISPR/Cas9 editing efficiency in sorghum (*Sorghum ‘Bicolor’*(L.) Moench) [162]. In 2022, Liu et al. developed an efficient *Arabidopsis* γ-glutamylcysteine synthase promoter, called AtGCSpro, with an average homozygous/double mutation frequency 1.7-fold and 8.3-fold higher than the p2 × 35Spro-Cas9 system for single and two target sites in the genome, respectively [163].

Germline-specific promoters for Cas9 expression can greatly improve the frequency and heritability of mutations in plants and avoid the creation of somatic chimeras, compared with constitutive promoters [164]. In 2015, Yan et al. used the *YAO* promoter to drive CRISPR/Cas9, which significantly enhanced the efficiency of Cas9 gene editing due to the preferential expression of the gene in tissues with active cell division [165]. In 2020, Ordon, J et al. used the DD45 and RPS5a promoters in the *Arabidopsis* genome to edit the system approximately 25–30-fold more efficiently compared with the previous ubiquitin promoter [166]. In 2020, Zheng et al. achieved high editing efficiency in soybean hairy root transformation using the oocyte-specific promoter AtEC1.2e1.1p [167]. In 2021, Kong et al. established a GLABRA2 mutation-based visible selection (GBVS) system to generate non-chimeric mutants in T_1_ generation *Arabidopsis* generated by the oocyte-specific CRISPR/Cas9 system. GBVS enhanced mutation screening overall, with a 2.58~7.50-fold increase in frequency, and 25~48.15% of T_1_ generation *Arabidopsis* screened by the GBVS system were homozygous or biallelic mutants, a 1.71~7.86-fold higher proportion than those screened using the original system [168]. In 2022, Wang et al. used the *pYAO* promoter to drive CRISPR/Cas9 to generate 45.83% homozygous single mutations in the *MePDS* gene, opening a more functional pathway for the genetic improvement of cassava SC8 [169]. The use of a germline-specific promoter would have multiple benefits, such as reducing the potential toxicity associated with Cas9 expression under other strong constitutive promoters. In addition, Cas9 expression in germ cells (oocytes, daughter cells, and early embryos) leads to heritable editing and reduces somatic mutations in organogenic plants. Efficient CRISPR/Cas9 systems based on germline-specific promoters may reduce chimerism and thus reduce the workload for the characterization of edited plants [170,171,172].

### 3.4. Other Strategies and Methods to Improve the Efficiency of CRISPR/Cas Family Editing

CRISPR/Cas9 gene-editing technology is a powerful tool for introducing specific mutations in organisms, including plants, and has recently been widely disseminated in molecular breeding. Such widespread use has also exposed many factors that affect the efficiency of gene editing, and scientists have conducted numerous investigations into optimizing CRISPR/Cas9 gene-editing technology.

The addition of elements with different functions for different promoters and target genes in the editing vector has the effect of increasing editing efficiency. In 2018, Mao et al. found that silencing AGO1 in tomatoes by introducing an AGO1-RNAi cassette into a CRISPR/Cas9 vector could improve gene-editing efficiency, thereby demonstrating that suppressing RNAi in plants can improve editing efficiency [173]. In 2019, Wang et al. added an Amir-RDR6 to CAMV35S-driven Cas9 and successfully suppressed endogenous RDR6 levels to improve the efficiency of CAMV35S-driven CRISPR/Cas9 gene editing [174].

Virus-induced genome editing (VIGE) systems are designed to induce targeted mutations in seeds without any tissue culture. Recently, gRNA systems delivered by viruses have been increasingly developed due to their simplicity and efficiency. In 2019, Hu et al. designed a barley streak mosaic virus (BSMV)-based gRNA delivery system that achieved positive target mutagenesis in the *TaGASR7* and *ZmTMS5* genes of wheat and maize with 78% and 48% efficiency [175]. In 2021, Kong et al. edited germ cells of wild tobacco using tobacco rattler virus (TRV): haploid mutations occurred in three target genes in tobacco seeds using the gRNA delivered by TRV [154].

The process of screening gene-editing plants requires a great deal of efficiency and time, and providing visible markers to detect the presence of transgenes has, likewise, recently been a key strategy for optimizing CRISPR/Cas9 gene-editing technology. The GFP fluorescent marker is a commonly used screening marker. In 2018, Tang et al. tested the ability of a fluorescent tag driven by a constitutive 35S promoter to adequately identify transgene-free mutants in dicots including *Arabidopsis*, European oilseed rape (*Brassica napus* L.), strawberry (*Fragaria × ananassa* Duch.), and soybean [176]. In 2019, Petersen et al. added a GFP fluorescent marker to Ben’s tobacco transformation vector and used fluorescence-activated cell sorting (FACS) to improve protoplast CRISPR/Cas9 editing efficiency 3~5-fold [177]. In 2020, He et al. coupled a CRISPR/Cas9 cassette with a unit that activated anthocyanin biosynthesis to provide a screening marker for visibility. In 2022, Trinh et al. screened transgenic hairy roots by adding a GFP fluorescent marker to the editing vector [178].

Temperature is also a factor in the efficiency of gene editing. In 2020 Milner et al. found that increasing the temperature of tissue culture or seed germination and early growth periods increased the frequency of mutations in wheat when the Cas9 enzyme was driven by the *ZmUbi* promoter rather than Osactin [179].

SgRNA length has been another key factor explored by researchers in recent years. In 2022, Liu et al. explored the effect of different sgRNA lengths on the editing efficiency of the rice genome and found a normal distribution of editing efficiency and sgRNA length, with 20 nt sgRNA (25%) being the most efficient. The editing efficiency decreased slightly with 1-2 bases (19 nt 20%, 18 nt 21%) but significantly with three bases (17 nt 4.5%) [180].

In addition to the widely used Cas9, Cas12a is often used for multiple gene editing in plant genomes [181]. However, the CRISPR/Cas12a system shows different editing efficiencies at genomic loci and is significantly affected by CRISPR RNA (crRNA) [182,183,184]. To improve the efficiency of the CRISPR/Cas12a system for multiple genome editing, researchers have conducted very intensive investigations. Optimization of crRNA expression is one strategy to improve the efficiency of Cas12a editing. In 2020, Hu et al. used modified tRNA-crRNA arrays to not only effectively achieve multiple genome editing in rice, but also to successfully edit target loci that were not edited by crRNA arrays, improving the efficiency of gene editing [185]. In 2021, William et al. performed multiple genome editing in *Arabidopsis* using *Moraxella boroculi* 3 Cas12a and successfully achieved the expression of a single transcript with as many as 13 crRNAs [186]. Temperature is also an essential factor in the efficiency of Cas12a editing. In 2020, An et al. used AsCas12a in poplar genome editing and optimized the co-culture temperature after *Agrobacterium*-mediated transformation from 22 °C to 28 °C, improving the efficiency of poplar Cas12a nuclease editing to 70% [187]. LbCas12a (*Lachnospiraceae* bacterium ND2006) is the most used Cas12 variant enzyme in plant molecular breeding. In 2021, Wang et al. used a guide consisting of the RNA polymerase II promoter and ribonuclease from switchgrass to bind to LbCas12a for eightfold more efficient gene editing in the wheat genome [188]. In 2022, Errin et al. reported the integration of HUH nucleic acid endonuclease with LbCas12a to increase the rate of targeted integration of donor DNA in plants to 26%, a fourfold increase compared with the control [189]. Cas12a is also widely used in plant molecular breeding, often for multiple editing of multiple target genes; improving the efficiency of Cas12a gene editing has also advanced the CRISPR family for use in plant molecular breeding [190,191,192,193,194].

## 4. Discussion and Perspective

The plant genome editing revolution offers many opportunities for functional genetics research and crop breeding. CRISPR/Cas9 has emerged as a revolutionary tool for efficiently targeted transgenesis and for inventing new CRISPR-based editing tools to achieve different goals of genome engineering, such as improved yields, pathogen resistance, improved nutritional efficiency, and abiotic tolerance of crop species. In recent years, the use of CRISPR/Cas9 to produce transgene-free plants with desired agronomic traits, and without the introduction of any exogenous DNA, has been widely reported, thus dispensing with the definition and regulation of GMOs [195]. With its high target programmability, specificity, and simplicity, CRISPR/Cas9 enables precise genetic manipulation of crop species, offering the opportunity to create germplasm resources with beneficial traits and to develop novel, more sustainable agricultural systems. Although CRISPR/Cas9 or CRISPR/Cas9-based technologies have made significant progress in the last few years, there is still room for applications to be explored and improved. Improving the efficiency of gene editing by CRISPR can be of great help in creating more dominant plant traits and obtaining desirable gene-editing positive plants. This review described a variety of strategies and methods for optimizing CRISPR/Cas9 gene editing systems. Using multiple strategies and crossovers in applications often leads to better results, as seen in 2019, when Wu et al. used controlled sgRNA length and engineered Cas9-*pmCDA1* to achieve high editing efficiency in rice [112].

Since 2020, dCas9 and nCas9-based base editors (BEs) and prime editors (PEs) have been widely used to enable precision editing in plant genomes [196,197,198]. The base editor and prime editor produce the desired changes in the genome without the introduction of donor DNA and DSBs and are 10 to 100 times more efficient at gene editing than HDR. BEs and PEs can certainly contribute to the development of superior varieties with increased yields, improved nutrient content, broad adaptability in the environment, and increased efficiency in the use of agricultural inputs [199,200,201]. In 2022, Li et al. achieved an editing efficiency of up to 66.7% using PE for precision editing at two adjacent target loci within the rice waxy gene [202]. In 2023, Qiao et al. achieved simultaneous genetic precision editing of multiple genes and targets in maize with an optimized PE system [114]. In 2023, Gaillochet et al. used ITER to develop optimized LbCas12a-ABE to obtain base-edited plants in stable wheat transformants with up to 55% editing efficiency and the ability to pass the edits on to T_1_ progeny [203].

In 2023, Jacobsen et al. discovered that CasΦ is capable of producing stable genetic editing in the model plant *Arabidopsis*. The CasΦ protein comprises only 700-800 amino acids, much smaller than Cas9 (1000–1400 aa) and Cas12a (1100–1300 aa) [204]. In addition, the characteristic of CasΦ to recognize T-rich PAM motifs and its adaptability to working temperatures due to the wide distribution of jumbo phages in various ecosystems make it an excellent potential application [205,206]. In 2023, Jennifer et al. developed a gene-editing targeted modulation system called CRISPR/Csm based on the type III CRISPR/Cas system, which is more efficient and less off-target. The CRISPR/Csm system does not require any PAM for target selection, and a 32 nt crRNA produces the highest knockdown efficiency [207,208]. In the future, increasingly convenient, advanced, and efficient CRISPR systems will be used in the field of plant molecular breeding, helping researchers create new and better plant traits.

## Figures and Tables

**Figure 1 plants-12-01478-f001:**
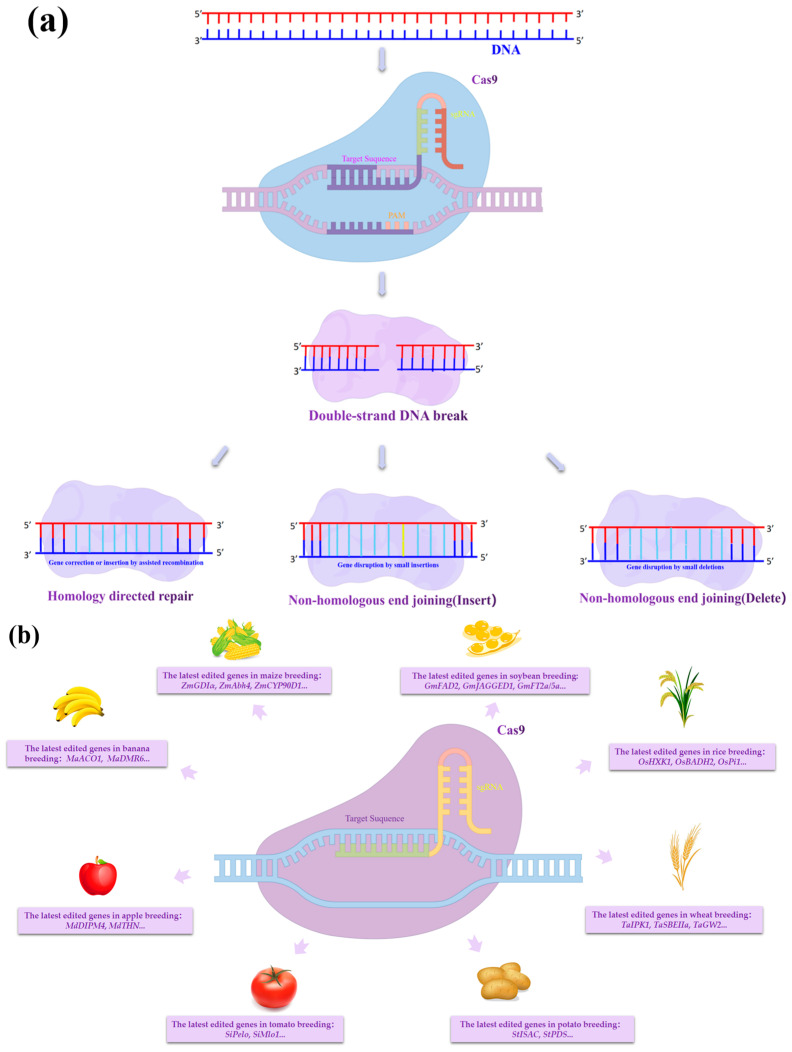
Principle of CRISPR/Cas9 gene-editing technology and targeting genome: (**a**) Mechanism of CRISPR/Cas9-mediated DNA repair; (**b**) CRISPR/Cas9 targets different plant genomes. CRISPR/Cas9 targeting of the target genome first produces a double-stranded DNA break, followed by two DNA repair mechanisms, non-homologous end-joining, and homologous recombination, in which editing occurs during the repair process. Non-homologous end-joining often produces deletions and insertions of target sequences, and homologous recombination produces exchanges of target sequence fragments or base substitutions. The figure mentions the names of the genes recently targeted by CRISPR/Cas9 in different major crops.

## Data Availability

The data is contained within the manuscript.

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
