# Peer review of "Strategies and Methods for Improving the Efficiency of CRISPR/Cas9 Gene Editing in Plant Molecular Breeding"

_plants, 2023, doi:10.3390/plants12071478_

Round 1

Reviewer 1 Report

Dear Authors,

thank you for well prepared and highly professional manuscript, dealing with a very topical subject. The manuscript requires only a few formal corrections.

The full names of the species, including authorship, must be given (rows 330, 331, 337, 413 etc.)

I suggest to include the latins names of species in table 2.

The labels of Tables 1 and 2 are incorrect. Not "Table.1." but "Table 1".

Please, provide more detailed description and explanation of the Fig. 1 under the figure itself. It would also be necessary to give explanations of abbreviations or designations.

Author Response

Please see the attached document for more details.

Reviewer 2 Report

The review of Zhou et al. entitled “Strategies and methods for improving the efficiency of 2 CRISPR/Cas9 gene editing in plant molecular breeding” aims at summarizing the progresses in this research field which hold great promises in generating plants with improved traits. The structure of the review is OK, as well as the information provided in tables. Unfortunately, in many instances the information provided is not useful, incomplete, or redundant, many sentences are poorly written, lacking clarity, Figure 1 is hard to understand (significance, letter size), and so on.

It is very unusual the way the authors describe some of the discoveries. For example: Line 25 “In 2002, the Jansen laboratory discovered …”, or line 412, “In 2023, Steven E Jacobsen and Jennifer A Doudna developed that CasΦ is capable ….” or line 417 “In 2023, Jennifer Doudna developed a gene editing targeted modulation system”. If the work was conducted in the laboratory of a researcher, it can be stated, but on cannot say “the Jansen laboratory”! or “Steven E Jacobsen and Jennifer A Doudna developed that”.

Same, why indicating that “people call this system the CRISPR/Cas system”? Why people? (line 29)

Other sentences (just a few examples) in which the writing style is hard understand, or it is wrong, are:

Lines 38-40

“Compared to conventional zinc-finger nucleases (Zinc-finger nucleases, ZFNs) and transcription activator-like effector nucleases (Transcription activator-like effector nucleases, TALENs), CRISPR/Cas9 gene editing technology has strong advantages in terms of 40 gene editing capabilities and convenience.” Why are the zinc-finger nucleases and transcription activator-like effector nucleases written twice?

Lines 72-75:

“Following the successful application of the CRISPR/Cas9 system in human cells, researchers successfully applied it to a variety of model crops including Arabidopsis thaliana and Nicotiana benthamiana, where the low efficiency of editing as quickly as possible but the generation of new plant traits allowed researchers to see its rich application.” Please rewrite and make this sentence shorter. It lacks clarity.

Lines 79-81

“In 2013, Shan et al. successfully used CRISPR/Cas9 gene editing technology to target the knockdown of AtPDS3 and NbPDS genes in Arabidopsis and tobacco, achieving the first application of CRISPR/Cas9 technology in plant genomes[45].”

If mentioning 2 plants and 2 genes, one have to add “respectively” after tobacco. In addition, one should mention the function of the genes, or what has been achieved by doing the knockdown.

Lines 81-83

“In 2013, Nekrasov et 81 al. used RNA-guided nucleic acid endonucleases in Cas9 to target the knockdown of the genome of the model plant Nicotiana benthamiana[46].”

This sentence makes no sense! How to use RNA-guided nucleic acid endonucleases in Cas9? How one can “target the knockdown of the genome”? Actually, can this technique knockdown a genome or only a gene or multiple genes?

Line 90

What is the difference between major crops and model crop? Actually what is a model crop?

Lines114-115

“Not only has multiple gene editing been achieved on a large scale in major crops, but surprising breeding results have been achieved in tomatoes, apples, bananas, and potatoes[67-72].” Why surprising?

Lines 57-60 and 116-120 are very similar like info provided. Should keep only one of them.

Line 206-208. This is quite a funny sentence (I apologize for writing this, but indeed is something special).

“In recent years, the use of researchers expressing multiple gRNAs targeting multiple target genes in plant editing vectors or expressing a single gRNA recognition site containing two or more target genes has been increasingly reported.”

Really, “the use of researchers expressing multiple gRNAs”?

Line 416-417

“wide distribution of macrophages”.

A macrophage is a type of white blood cell. Use giant, or large phages to avoid confusion.

Line 423

“to the field of plant molecular breeding, bringing us even greater surprises.”

Please change this last sentence of the review, do not end with “even greater surprises”. Maybe new discoveries, achievements or something similar.

Formatting issues (just a few examples)

Line 10

Change t with T.

Throughout the text. Why there is no space between the last word and the bracket of the citation? For example, “CRISPR/Cas system[1-3].” should be “CRISPR/Cas system [1-3].”

References

All species names in the references should be italicized. For example, see ref. 2, 46, 47, and so on.

While journal names are usually well formatted, the journal INTERNATIONAL JOURNAL OF MOLECULAR SCIENCES is always written with capital letters. Why? See ref. 20, 54, 63, 68 and so on.

Ref 31. CRISPR?Cas9 should be CRISPR/Cas9

Ref. 36 “ Journal of RNAi and gene silencing : an international journal of RNA and gene targeting research” should be abbreviated

Author Response

(The authors gave the same response as above.)

Round 2

Reviewer 2 Report

The revised version of the review entitled “Strategies and methods for improving the efficiency of 2 CRISPR/Cas9 gene editing in plant molecular breeding” addressed some of the issues mentioned before. However, in many instances, the authors failed to provide correct, or unbiased information. In a review it is crucial to cite the original work, not other reviews or papers in which authors are mentioning the work or method. By doing this, it can be considered as improper citation.

Also, in many places in the revised version of the manuscript the authors added, in addition to the common name of species, the scientific name. However, they are not using the common way to do it. For example, when indicating that maize is Zea mays L, for the first time one can write “maize (Zea mays L)”. I am not sure if it is needed to indicate the authority that described the species, in this case Linneus. The second time one can use either just “maize” (preferably) or “maize (Z. mays)”. It is not OK to write it throughout the text “maize (Zea mays L)”. They made these unusual changes in lines 90 to 119 and Table 2. Please revise the issue, as it relates to basic scientific writing.

Mistakes were also made, for example, in line 118 it is mentioned “tomatoes (Solanum tuberosum L.). This in the scientific name for potatoes, not tomatoes. Also, the scientific name for cultivated bananas is not Musa nana (the dwarf banana). This is just one of the many banana species that are cultivated. Many other are eaten, for example, Musa × paradisiaca, Musa acuminata (=Musa nana) and Musa balbisiana. Maybe just use Musa sp.

Major comments:

Line 25

In English (language and literature) when referring to an achievement, work carried out or a discovery coming from somebody’s lab on can say something like that person’s lab, or researcher working in that person’s lab. So instead of saying “Jason's scientific research team” one can say researchers from Jason’s lab or something similar. “Jason's scientific research team” sounds awkward.

Or better, in “In 2002, bioinformatics analyses led to the discovery of a novel family of DNA sequences found only in bacteria and archaea (1) 1 corresponds to Jansen et al. Mol. Microbiol. 2002, 43, 1565-1575, doi:10.1046/j.1365-2958.2002.02839.x.

The same situation can be found in lines 423-429. “In 2023, Steven E Jacobsen and Jennifer A Doudna’s scientific research teams” (line 423) and “In 2023, Jennifer Doudna's scientific research team developed” (line 429). Why mentioning the full names of researchers?

Everywhere else in the review, the mentioning the work of research group was done in a proper way, for example, line 420, “In 2023, Gaillochet et al.”. And again, “scientific research team” sounds odd in this context.

Lines 26-29

“They called this sequence ‘clustered regularly interspaced short palindromic repeats’ (CRISPR) and named the genes close to the CRISPR locus ‘Cas’ (CRISPR-associated); the scientist Zhang Feng named it the ‘CRISPR/CAS gene editing system’ [1-3].”

In this sentence the authors provided false information and wrong (not supporting) citation is provided. Also, they touch a very sensitive issue around the Nobel prize in Chemistry awarded in 2022. The information provided in this sentence indicate an unacceptable bias.

First of all, citation 1-3 are not related to the work of Dr. Zhang Feng.

Secondly, in the abstract of the 2002 paper of Jansen et al (Mol. Microbiol. 2002, 43, 1565-1575, doi:10.1046/j.1365-2958.2002.02839.x.) it is clearly stated that: “To appreciate their characteristic structure, we will refer to this family as the clustered regularly interspaced short palindromic repeats (CRISPR).”

So CRISPR term was coined by Jansen et al.

In 2022 the Noble prize for Chemistry was awarded jointly to Emmanuelle Charpentier of he Max Planck Unit for the Science of Pathogens and Jennifer Doudna of the University of California, Berkeley, "for the development of a method for genome editing." The two scientists were the first to show that CRISPR system could be used to edit DNA in an in vitro system (see Jinek et al., Science, 28 Jun 2012, vol 337, issue 6096, pp. 816-821, DOI: 10.1126/science.1225829).

In the abstract of the 2012 paper one can read: “Clustered regularly interspaced short palindromic repeats (CRISPR)/CRISPR-associated (Cas) systems provide bacteria and archaea with adaptive immunity against viruses and plasmids ….”.

Feng Zhang, working at the Broad Institute, showed 6 months later that CRISPR worked in mammalian cells. As these discoveries were done almost simultaneously, it was debated why Feng Zhang was not also nominated for the Nobel prize. The authors should indicate if indeed “the scientist Zhang Feng named it the ‘CRISPR/CAS gene editing system” and not other scientists before them. Also, by not indicating the paper of Jinek et al. from 2012, the authors show a very strong bias in acknowledging the contribution of outstanding scientists that were awarded the Nobel prize.

Line 29-31

“In 2013, researchers discovered that the CRISPR/Cas9 system relies on Cas9 proteins and small RNAs to detect precisely, and silence, foreign sequences for precise identification and editing of the genome.”

The authors mentioned Jansen’s lab as discovering CRISPR but they don’t mention that the researchers who discovered that the CRISPR systems relies on Cas9 were rewarded a Nobel prize for this major discovery! Very debatable approach! Please rewrite and provide a citation!

Line 40.

Flavobacterium okeanokoites should be italicized.

Line 45

In the CRISPR/Cas9 system, the Cas nuclease induces a double-strand breakage (DSB) at the designated target site of the gRNA [21].” Citation 21 is wrong. The fact that Cas nuclease induces a double-strand breakage (DSB) at the designated target site of the gRNA was described in 2013, in several publications, and not in 2018!

 Line 70

Citation is needed after human cells.

“Following the successful application of the CRISPR/Cas9 system in human cells,”

Figure 1 I still not OK. The quality of the figure is not good, and the drawing is oversimplified and not relevant. Also, one cannot read the names of the genes, and grapes (Vitis vinifera) are not indicated in the text as being edited.

There are more issues throughout the MS, so it is recommended that the authors have another look at the review and fix them.

Round 3

Reviewer 2 Report

The MS was improved. However, I have the following additional suggestions:

Lines 28-32

“In 2012, the working principle of CRISPR/Cas9 gene editing technology was successfully elucidated and was first applied in 2013 [2-3].”

How come it was elucidated in 2012 (and published in 2012) but applied only in 2013. It was validated in 2012!

Also, citation is confusing as the papers are from 2012 and 2013. Citation 3 should be used in the next sentence as it is related to multiplex editing!

I suggest a few minor changes for added clarity:

“In 2012, the working principle of 28 CRISPR/Cas9 gene editing technology was successfully elucidated [2]. In 2013, CRISPR/Cas9 was used for the first time in several fields to achieve not only a significant increase in gene knockout efficiency, but also to enable multiple gene knockouts [3-7].”

Line 85

Italicize Oryza sativa

Lins 134

The sentence “The figure briefly describes the names of the different plant genomes targeted by CRISPR/Cas9.” needs to be revised.

Please change it to something like: “The figure mentions the names of the genes recently targeted by CRISPR/Cas9 in different major crops.”
